# Review of T Helper 2-Type Inflammatory Diseases Following Immune Checkpoint Inhibitor Treatment

**DOI:** 10.3390/biomedicines12081886

**Published:** 2024-08-19

**Authors:** Yoshihito Mima, Tsutomu Ohtsuka, Ippei Ebato, Yukihiro Nakata, Akihiro Tsujita, Yoshimasa Nakazato, Yuta Norimatsu

**Affiliations:** 1Department of Dermatology, Tokyo Metropolitan Police Hospital, Tokyo 164-8541, Japan; 2Department of Dermatology, International University of Health and Welfare Hospital, Tochigi 324-8501, Japan; 3Department of Respiratory Medicine, International University of Health and Welfare Hospital, Tochigi 324-8501, Japan; 4Department of Diagnostic Pathology, International University of Health and Welfare Hospital, Tochigi 324-8501, Japan; 5Department of Dermatology, International University of Health and Welfare Narita Hospital, Chiba 286-0124, Japan; norimanorima@gmail.com

**Keywords:** immune checkpoint, immune checkpoint inhibitor, nivolumab, lung adenocarcinoma, prurigo nodularis, T helper-2 inflammation, PD-1, PD-L1

## Abstract

Immune checkpoints are mechanisms that allow cancer cells to evade immune surveillance and avoid destruction by the body’s immune system. Tumor cells exploit immune checkpoint proteins to inhibit T cell activation, thus enhancing their resistance to immune attacks. Immune checkpoint inhibitors, like nivolumab, work by reactivating these suppressed T cells to target cancer cells. However, this reactivation can disrupt immune balance and cause immune-related adverse events. This report presents a rare case of prurigo nodularis that developed six months after administering nivolumab for lung adenocarcinoma. While immune-related adverse events are commonly linked to T helper-1- or T helper-17-type inflammations, T helper-2-type inflammatory reactions, as observed in our case, are unusual. The PD-1–PD-L1 pathway is typically associated with T helper-1 and 17 responses, whereas the PD-1–PD-L2 pathway is linked to T helper-2 responses. Inhibition of PD-1 can enhance PD-L1 functions, potentially shifting the immune response towards T helper-1 and 17 types, but it may also influence T helper-2-type inflammation. This study reviews T helper-2-type inflammatory diseases emerging from immune checkpoint inhibitor treatment, highlighting the novelty of our findings.

## 1. Introduction

The tumor microenvironment (TME) is inherently immunosuppressive due to the coordinated actions of tumor cells and their surrounding stroma, which collectively create conditions favorable for tumor progression and immune evasion [1,2,3,4,5]. Tumor cells frequently overexpress immune checkpoint molecules, leading to the inhibition of T cell activation and a subsequent reduction in immune responses [1]. Additionally, the accumulation of immunosuppressive cells, such as regulatory T cells (Tregs) and myeloid-derived suppressor cells (MDSCs) within the TME further impedes effective immune responses against tumor cells [2]. Tumor cells also secrete immunosuppressive cytokines, including IL (IL)-10 and transforming growth factor-beta (TGF-β), which further inhibit T cell function and weaken anti-tumor immunity [3]. The TME is often marked by hypoxia and acidic conditions, which additionally compromise immune cell function [4]. Furthermore, tumor cells can downregulate the expression of major histocompatibility complex (MHC) class I molecules, making it more challenging for T cells to recognize and attack tumor antigens [5]. These factors collectively establish an immunosuppressive milieu that allows tumors to evade immune surveillance [1,2,3,4,5]. Additionally, tumor cells can send immunosuppressive signals to the bone marrow and spleen, which are crucial for the generation and maintenance of Tregs and the production of cytokines and chemokines, thereby perpetuating an immunosuppressive state [2,6,7]. The intricate interplay between tumor cells, the TME, and these hematopoietic organs ensures the sustenance of an immunosuppressive environment that supports tumor growth [2,6,7]. Consequently, immune checkpoints—central to maintaining this immunosuppressive balance by inhibiting T cell activation—are a primary focus of therapeutic strategies [1].

### 1.1. Programmed Death 1, Programmed Death Ligand 1 and Cytotoxic T-lymphocyte Associated Antigen 4

The immune checkpoints identified to date include primarily programmed death 1 (PD-1) and programmed death ligand 1 (PD-L1), cytotoxic T-lymphocyte associated antigen 4 (CTLA-4), and lymphocyte activation gene 3 (LAG-3), among others [8]. PD-1 belongs to the CD28 family and is a co-inhibitory transmembrane protein expressed on antigen-stimulated T lymphocytes and B lymphocytes, as well as natural killer cells, myeloid-derived suppressor cells, and various other cells [9]. When PD-1 binds to its corresponding ligand, it can diminish the response of T cells to T cell receptor (TCR) stimulation signals and modulate the strength of the immune response [9]. Most studies on PD-1 ligands focus on PD-L1 due to a lack of research discussing the role of PD-L2 in tumor immune systems [10]. PD-L1 is expressed by tumor cells, epithelial cells, dendritic cells, macrophages, fibroblasts, and exhausted T cells, and its expression intensity is influenced by cytokines such as interferon-gamma (IFN-γ) and carcinogenic factors [10,11]. When PD-L1 binds to PD-1 and inhibits various signaling pathways, the proliferation and differentiation of effector T cells are inhibited [10,11].

CTLA-4 is a type I transmembrane glycoprotein belonging to the immunoglobulin superfamily. It is abundantly expressed in tumor tissues and is frequently found in the cytoplasm of CD4+ and CD8+ T cells [12]. CTLA-4 is recognized as a negative regulator of anti-tumor immunity [8]. Upon induction on the cell surface, it binds to CD80 and CD86 on antigen-presenting cells (APCs) with a higher affinity than the co-stimulatory molecule CD28 on T cells. This inhibits cytotoxic T cell activity, enhances Tregs immunosuppressive activity, and enables immune avoidance by tumor cells [12]. Other immune checkpoints include LAG-3, T cell immunoglobulin and mucin-domain containing 3 (TIM-3), CD47, T cell immunoreceptor with Ig and ITIM domains (TIGIT), and V-domain Ig suppressor of T cell activation (VISTA) [8].

Immune checkpoint inhibitors (ICIs) are designed to disrupt the regulatory pathways that inherently limit T cell activity, thereby inducing intrinsic regulation of the activation and maintenance of T cell function [13]. Detailed studies on the mechanisms of immune checkpoints have shown that ICIs such as CTLA-4 and PD-1/PD-L1 exhibit effective anti-tumor activity in malignancies such as urothelial carcinoma, renal cell carcinoma, Hodgkin lymphoma, non-small-cell lung cancer, and melanoma. This has led to their wide usage in clinical practice [14]. Monoclonal antibodies targeting PD-1/PD-L1, including nivolumab, pembrolizumab, cemiplimab, durvalumab, avelumab, and atezolizumab, as well as monoclonal antibodies targeting CTLA-4, such as ipilimumab and tremelimumab, are commonly used in clinical practice [14].

### 1.2. Immune-Related Adverse Events

ICIs disrupt the immune balance, reducing T cell tolerance and leading to the development of immune-related adverse events (irAEs) [15]. However, the rate of irAE onset is very high, affecting 90% of patients treated with anti-CTLA-4 and 70% of patients treated with anti-PD-1/PD-L1 therapies [16]. The incidence of drug adverse effects is higher with anti-CTLA-4 therapy compared to anti-PD-1 and anti-PD-L1 therapies, and combination therapy is linked to a higher incidence of adverse events in comparison with monotherapy [17]. IrAEs are organ-specific, with skin-related irAEs being the most common, typically presenting as slight itching or eruptions. Following skin-related issues, gastrointestinal toxicities are frequently observed, often manifesting as diarrhea or colitis [18]. The third most common irAEs are endocrine dysfunctions, such as thyroid dysfunction, including both hypothyroidism and hyperthyroidism, adrenal insufficiency, and pituitary inflammation. Musculoskeletal toxicities, such as arthralgia and myalgia, and ocular toxicities, such as uveitis and dry eye syndrome, are also commonly reported [16]. Additionally, while the onset of pneumonia, myositis, neurotoxicity, myocarditis, and hematologic toxicities are very rare, their development is noteworthy due to their potential severity [16]. Notably, CTLA-4 inhibitors are more likely to cause colitis, hypophysitis, and rash, whereas PD-1/PD-L1 inhibitors are more likely to cause pneumonitis and thyroid dysfunction. Overall, the most frequent adverse effects related to the combination of CTLA-4 and PD-1/PD-L1 inhibitors are cutaneous and endocrine adverse effects [19].

The precise pathophysiological mechanisms of irAEs remain unclear; however, they share various similarities with several autoimmune diseases. IrAEs are thought to be primarily associated with the disruption of self-tolerance and an increased sensitivity of the body to antigen recognition, leading to the attack of self-tissues [20]. Notably, the production of autoantibodies, T cell infiltration, and the mediation of inflammatory cytokines, such as various interleukins, are considered to lead to the onset of irAEs [20]. Typically, the normal function of immune regulation within the body maintains the appropriate balance of immune activation and immune tolerance through co-stimulatory pathways of reactive T cells. In contrast, immune tolerance suppresses the activation of self-reactive T cells and regulates the strength of the immune response [21]. In this context, regulatory co-stimulatory molecules on naive T cells can control T cell activation, tolerance, and immune-mediated tissue damage by binding to T cell ligands [21]. Thus, it is believed that ICIs lead to the disruption of self-tolerance and increased body sensitivity to antigen recognition through ICI-induced T cell activation. This disturbance of the immune balance causes the body to attack its own tissues.

### 1.3. Activation of CD4- and CD8-Positive T Cells and Suppression of Tregs Following Immune Checkpoint Inhibitor Treatment

ICIs can activate both CD4-positive T cells and CD8-positive T cells [22,23,24,25,26,27]. CD4 T cells, particularly Th1 and Th17 cells, are primarily involved in several irAEs, including colitis, liver dysfunction, pneumonia, and skin disorders [23,24]. On the other hand, the activation of CD8-positive T cells is implicated in irAEs such as myocarditis, type 1 diabetes, and psoriasis [25,26,27]. Furthermore, ICIs inhibit immune checkpoint molecules, preventing the immune avoidance of tumor cells. This disruption of peripheral T cell tolerance leads to the accelerated diversification and clonal multiplication of cytotoxic cells, resulting in high inflammation and autoimmunity [16,28]. Notably, both CTLA-4 inhibitors and PD-1 inhibitors enhance T cell activation and proliferation, potentially attacking both malignant and non-malignant tissues by eliminating the functions of Tregs which maintain immune tolerance [25]. Therefore, ICIs induce irAEs by acting on CD4-positive T cells, CD8-positive T cells, and Tregs [16,22,23,24,25,26,27,28]. 

### 1.4. Production of Autoantibodies Following Immune Checkpoint Inhibitor Treatment

Moreover, ICIs can also activate autoreactive B cells and increase the production of autoantibodies. This triggers the classical complement cascade reaction when these antibodies bind to target antigens [22]. This effect is illustrated by case studies showing that some patients who were negative for several antibodies before ICI treatment later developed autoimmune antibodies, such as anti-thyroglobulin antibodies, after starting ICI treatment [22].

### 1.5. Production of Cytokines Following Immune Checkpoint Inhibitor Treatment

The levels of various cytokines, including interleukins, tumor necrosis factor, and interferons, can significantly change before and after initiating ICI treatment. These cytokines may contribute to the development of immune-related adverse events [23]. Inflammatory mediators released by immune cells can cause immune-mediated damage in anatomically susceptible tissues, suggesting that the severity of tissue-specific or general cytokine production may contribute to the pathogenesis of irAEs. These cytokines bind to immune cells, activating intracellular signaling pathways and leading to dysregulation of the inflammatory response [23].

### 1.6. Cutaneous Immune-Related Adverse Events

It is believed that multiple factors, such as T cell infiltration, B-cell-related autoantibody production, and inflammatory cytokine mediation, are intricately involved in the development of organ-specific irAEs, including cutaneous irAEs [16,22,23,24,25,26,27,28]. For example, two-thirds of patients with skin irAEs require oral prednisone therapy to treat rashes, and 19% of patients stop taking ICIs due to these side effects [29]. Mild to moderate skin irAEs often include itching, a non-specific rash called maculopapular rash, a rash resembling lichen planus, psoriasis-like dermatitis, eczema, vitiligo, and hair loss [30]. Severe skin irAEs have also been reported, such as bullous pemphigoid (BP), Stevens–Johnson syndrome, toxic epidermal necrolysis, and drug-induced hypersensitivity syndrome [30].

### 1.7. T Helper 1, 2, and 17 Immune Responses Following Immune Checkpoint Inhibitors Treatment

Having covered an overview of ICIs and their adverse effects, the focus now shifts to examining how T helper (Th)1, Th2, and Th17 cytokines lead to the occurrence of different irAEs. PD-1, a member of the immunoglobulin superfamily, functions as an inhibitory receptor found in activated T cells and B cells that helps to maintain peripheral tolerance [31]. PD-1 interacts with two ligands, PD-L1 and PD-L2, present not only in APCs but also in peripheral tissues. PD-L1 is found in various hematopoietic and non-hematopoietic cells, while PD-L2 is mainly found in APCs such as macrophages, dendritic cells, and some B cells [32,33,34]. Notably, PD-L2 binds to PD-1 with greater affinity than PD-L1 [35]. Recent investigations have indicated that APCs express PD-L1 in response to IFN-γ or IL-17A stimulation, while PD-L2 expression is driven by IL-4 stimulation [36]. Additionally, mouse model experiments focusing on Th1-mediated contact hypersensitivity, Th2-mediated atopic dermatitis (AD)-like inflammation, and Th17-mediated psoriasis-like dermatitis showed that PD-L1 knockout mice had accelerated Th1 and Th17 immune responses, whereas PD-L1 knockout mice demonstrated worse Th2 immune responses [36]. This suggests that PD-L1 plays a significant role in Th1 and Th17-related inflammation, while PD-L2 is involved in Th2-related inflammation [36]. Therefore, the PD-1–PD-L1 axis is suggested to control Th1 and Th17 immunity, which are target sof current immune checkpoint inhibitors, while the PD-1–PD-L2 axis regulates Th2 immunity [36]. Therefore, when the function of PD-1 is regulated, it can lead to the activation of Th1 and Th17 immunity, previously suppressed by the PD-1–PD-L1 axis, and it may also activate Th2 immunity, previously suppressed by the PD-1–PD-L2 axis.

In this way, PD-1 inhibitors can enhance immune responses through both the PD-1–PD-L1 and PD-1–PD-L2 axes, which may lead to skin inflammations driven by various types of Th1, Th2, and Th17 cytokines [31,32,33,34,35,36]. However, most of the reported skin diseases induced by PD-1 inhibitors so far are related to Th1 and Th17 cytokines. For instance, Reschke et al. reported that most cases of ICI-triggered dermatitis are predominantly Th1 cytokine-type inflammation [37]. Additionally, anti-PD-1 therapy is often reported to cause T-cell-mediated adverse skin effects, including Th17-mediated inflammatory diseases, such as psoriasis, and neutrophilic inflammatory diseases, such as hidradenitis suppurativa [38,39]. Moreover, studies using single-cell analysis have suggested that PD-1 inhibitors increase both Th1 and Th17 cells in clusters of CD4-positive T cells, thereby promoting Th1 and Th17-type inflammation. Overall, while there have been numerous reports regarding Th1 and Th17 inflammation induced by PD-1 inhibitors, there is a lack of literature on Th2-type inflammation caused by PD-1 inhibitors. 

In this study, we report a rare case of PD-1 inhibitors-induced prurigo nodularis (PN), which is a chronic inflammatory cutaneous disease and is strongly associated with type 2 inflammation [40]. Furthermore, we review previously reported cases of Th2-type inflammation induced by PD-1 inhibitors.

## 2. Case Presentation

A 51-year-old Japanese woman was diagnosed with stage IV lung adenocarcinoma with adrenal metastasis based on the results of computed tomography and biopsies of both tumors (Figure 1a,b).

Nivolumab, an anti-PD-1 receptor antibody, was initiated due to an elevated carcinoembryonic antigen level (658 mg/mL). After three years, the patient’s carcinoembryonic antigen level remarkably improved (2.3 mg/mL) and their lung and adrenal tumors had significantly regressed, leading to the discontinuation of nivolumab. The patient had a history of AD during childhood, which improved when they became an adult. Since then, the patient has not experienced any skin diseases, including psoriasis vulgaris and PN. In addition, no new medications were introduced following the diagnosis of malignancies, aside from nivolumab. However, six months after starting nivolumab, the patient developed itchy eruptions on their extremities. Although they received continued treatment with topical betamethasone butyrate propionate and oral antihistamines during the period of ICI administration, the eruptions were persistently refractory to these treatments. Even after discontinuing nivolumab, the widespread rash remained and was refractory to any ointments or oral antihistamines. Physical examination revealed multiple nodular prurigo on the extremities, some with erosions and ulcers in the center and some with hyperpigmentation (Figure 2).

Laboratory examination revealed elevated IgE (592 IU/mL) and TARC (1703 pg/mL). Markers for bullous diseases, such as anti-desmoglein 1 antibody, anti-desmoglein 3 antibody, and anti-bullous pemphigoid 180 antibody, were all negative. Histopathological examination from nodular prurigo on the extremities revealed epidermal thickening, liquefaction degeneration, spongiotic dermatitis in the epidermis, and inflammatory cellular infiltration around vessels in the dermis (Figure 3a,b).

Direct immunofluorescence did not show any deposits of immunoglobulin (Ig)G, IgA, IgM, C3, or C4. The clinical and histopathological findings led to the diagnosis of PN. Considering the absence of any history of skin diseases and the lack of introduction of any new medications after the diagnosis of malignant tumors, it was considered that the onset of PN was related to the initiation of nivolumab. The patient is currently being treated with topical betamethasone dipropionate and oral olopatadine (10 mg/day), but the nodular eruptions have shown little improvement.

## 3. Review

This report provides a comprehensive review of the Th2-type inflammatory diseases associated with ICI administration.

### 3.1. Eosinophilia

Immune-related eosinophilia is a new adverse event associated with anti-PD-1 or anti-PD-L1 treatments [41]. Th2 cells differentiate from CD4 helper T cells under the influence of IL-4 and conventional or monocyte-derived CD11b dendritic cells. Th2 cells produce cytokines such as IL-4, IL-5, and IL-13, which can induce immunoglobulin class switching to IgE [42]. Overexpression of IL-5 significantly increases eosinophil counts and antibody levels in vivo. IL-5 closely contributes to eosinophilic proliferation, survival, and maturation [43]. Therefore, eosinophilia is considered a Th2-type inflammation related to the activation of IL-5, one of the Th2-type cytokines [41,42,43]. In a study by Bernard-Tessier et al., 909 patients treated with anti-PD-1 or anti-PD-L1 therapy were observed, and 28 patients (2.9%) developed eosinophilia [44]. In patients with metastatic melanoma treated with pembrolizumab, an increase in eosinophilic counts during treatment has been shown to be a predictive biomarker for anti-tumor responses and overall survival [45]. Recent studies suggest that infiltrating eosinophils around tumors may contribute to tumor control by secreting chemoattractant cytokines that recruit CD8-positive T cells and macrophages, indicating that increased eosinophils may play a role in improving survival [45,46]. 

PD-1 inhibitors associated eosinophilia can be linked to eosinophilic pneumonia (EP), eosinophilic gastrointestinal disorders, eosinophilic bronchiolitis, allergic bronchopulmonary aspergillosis, eosinophilic granulomatosis with polyangiitis, asthma, eosinophilic fasciitis, and eosinophilic folliculitis (EF). These will be discussed in the following sections.

### 3.2. Eosinophilic Pneumonia

EP is a lung disease characterized by Th2 inflammation. It was first reported in 1989 and is identified by a significant increase in bronchoalveolar lavage eosinophils [47,48]. EP is associated with various factors, such as exposure to smoking, environmental factors, occupational dust, inhalation of toxins, and certain medications, including minocycline and cephalosporins [49]. Lung dendritic cells express high levels of PD-L2, which inhibits the response of Th2 cells expressing PD-1. Therefore, when the interaction between PD-1 and PD-L2 is inhibited by anti-PD-1 antibodies, Th2 inflammation may be promoted by removing the inhibition of Th2 inflammation [50,51,52]. 

IL-5, a Th2 cytokine, supports eosinophil production in the bone marrow. Th2 cytokines induced eosinophils facilitates their mobilization to the lung mucosa and stroma, leading to an increased presence of eosinophils in lung tissue. This process may contribute to the development of EP as an irAE [53]. So far, only two cases of EP have been reported following ICI administration [53,54]. Jodai et al. reported a case of EP in a 62-year-old man two months after nivolumab administration for lung adenocarcinoma [53]. Similarly, Mohammed et al. reported a case of EP following pembrolizumab and anti-TIGIT therapy for lung adenoid cystic carcinoma [54]. Both patients were treated with oral prednisone (30 mg) and intravenous prednisone, respectively, resulting in complete regression [53,54].

### 3.3. Eosinophilic Gastrointestinal Disorders (Enteritis, Esophagitis)

To date, there have been three reported cases of ICI-induced eosinophilic gastrointestinal disorders [55,56,57]. Yang et al. described a case of eosinophilic enteritis following ipilimumab and nivolumab administration for melanoma [55]. In addition, Barnett et al. reported a case of eosinophilic esophagitis following pembrolizumab administration for non-small-cell lung cancer [56]. Similarly, Nakamura et al. reported a case of eosinophilic gastrointestinal disorder following pembrolizumab administration for non-small-cell lung cancer [57]. Primary eosinophilic gastrointestinal disorders are rare diseases characterized by eosinophilic infiltration of the gastrointestinal tract without any other identifiable causes of intestinal eosinophilia. Notably, eosinophilic gastritis, enteritis, and gastroenteritis are often categorized together as the same group due to their clinical similarities [58]. Previous research involving transcriptome analysis of gastric biopsies from patients with eosinophilic gastroenteritis has revealed the activation of Th2 cytokine signaling pathways, including IL-4, IL-5, and IL-13 [59,60]. Thus, the molecular pathogenesis of eosinophilic gastroenteritis is believed to be mediated by a Th2-driven immune response that induces eosinophil chemotaxis and activation [59,60]. In all three cases of ICI-induced eosinophilic gastrointestinal disorders, high-dose oral corticosteroid therapy was administered without continuing chemotherapy, which eventually led to improvement [55,56,57].

### 3.4. Allergic Bronchopulmonary Aspergillosis

Allergic bronchopulmonary aspergillosis (ABPA) is believed to be caused by sensitization to *Aspergillus* and is closely associated with asthma [61]. ABPA is currently understood to be triggered by a robust Th2 cell response to the fungus, resulting in significant mucus production, hypersensitivity, bronchiectasis, mucus plugging, eosinophilia, and frequent exacerbations of asthma accompanied by elevated IgE levels [62]. Donato et al. reported a case of ABPA following pembrolizumab administration for lung squamous cell carcinoma [63]. After four months of treatment with pembrolizumab, the patient exhibited elevated IgE levels and an increase in *Aspergillus fumigatus*. The researchers suggested that the patient’s eosinophilic asthma background might have contributed to a hypersensitive reaction due to enhanced Th2 cytokine activity following PD-1 inhibitor administration, leading to the development of ABPA. Eventually, the patient stopped taking pembrolizumab and started treatment with the antifungal medication voriconazole and oral steroids. Eventually, their ABPA improved, and the patient could safely resume pembrolizumab therapy. This is the only reported case of ABPA induced by ICI administration, and more cases need to be accumulated to gain a better understanding of this potential adverse effect.

### 3.5. Eosinophilic Bronchiolitis

Eosinophilic bronchiolitis is a rare condition characterized by dyspnea, productive cough, and eosinophilia in the bronchioles. It often occurs alongside asthma and sinusitis [64,65]. Tamura et al. reported a case of eosinophilic bronchiolitis following treatment with atezolizumab, a PD-L1 inhibitor, for advanced breast cancer [66]. The patient’s symptoms improved rapidly after beginning high-dose oral corticosteroid therapy. Despite the inhibition of the PD-1–PD-L1 pathway, which is known to enhance Th1 and Th17 inflammation, this case developed eosinophilic bronchiolitis, a Th2 inflammation-related disease [36]. Th17 cells produce IL-17 and IL-23, with IL-17 interacting with eosinophils to trigger eosinophil degranulation [67] and IL-23 promoting eosinophil mobilization to the airways through Th2 inflammatory pathways [68]. When analyzing this case, Tamura et al. speculated that PD-L1 inhibition activates eosinophil functions via IL-17 and IL-23 cytokines, possibly leading to eosinophilic bronchiolitis [66,67,68]. However, although other eosinophil-related diseases have been associated with PD-1 inhibitors, this case is unique in its association with a PD-L1 inhibitor, and Tamura et al.’s report remains the only documented case of ICI-induced eosinophilic bronchiolitis. Thus, it is possible that there are other factors, aside from the PD-L1-mediated mechanisms involving IL-17 and IL-23 cytokines, that contribute to the development of eosinophilic bronchiolitis. Further research and accumulation of cases are needed to better understand the underlying mechanisms and contributing factors of eosinophilic bronchiolitis.

### 3.6. Eosinophilic Granulomatosis with Polyangiitis

Harada et al. reported a case of asthma and eosinophilic granulomatosis with polyangiitis (EGPA) following nivolumab administration for renal large cell neuroendocrine carcinoma, which showed significant eosinophil infiltration into the lung tissue [69]. The pathogenesis of EGPA primarily involves CD4-positive Th2 cells, B cells, and eosinophils. IL-5, in particular, is associated with the development of EGPA, as it plays a crucial role in eosinophil survival and function [70]. Mepolizumab, an IL-5 inhibitor, is an approved treatment for EGPA and was administered in the case reported by Wechsler et al. [71]. Notably, the authors suggested the possibility that the PD-1 inhibitor activated PD-L2 functions, leading to the overproduction of Th2-type cytokines and the onset of asthma and EGPA [69].

### 3.7. Asthma

Asthma is a prolonged inflammatory disease of the airways that is marked by airflow obstruction and airway hyperresponsiveness. The pathogenesis of asthma is diverse and intricate; however, eosinophilic inflammation remains a central pathological feature [72]. IL-5, a Th2 cytokine, plays a pivotal role as a hematopoietic cytokine that induces the differentiation, maturation, migration, and activation of eosinophils. Because eosinophil count is associated with severe asthma exacerbations and overall asthma control, IL-5 is closely related to asthma management [73]. In a case where asthma developed following pembrolizumab administration, increased levels of IL-5 cytokine in the blood were observed, suggesting a possible link between PD-1 inhibitor administration and the onset of asthma through IL-5 cytokine production [74]. Similarly, in a study involving asthma mouse models, PD-L2 was primarily expressed in dendritic cells, and both PD-1 and PD-L2 were upregulated, creating a condition conducive to the production of Th2 cytokines. Additionally, in vivo administration of PD-L2 in mouse asthma models led to increased serum IgE levels, eosinophil and lymphocyte infiltration into the bronchoalveolar lavage fluid, increased cell counts in draining lymph nodes, and production of IL-5 and IL-13 in tissues, indicating that PD-L2 may contribute to asthma exacerbation [75].

To date, six cases of newly developed asthma following ICI administration have been reported [74,76,77,78,79,80]. Theoretically, the administration of PD-1 inhibitors blocks the PD-1–PD-L2 pathway, enhancing PD-L2 function and leading to increased production of Th2 cytokines [36], which could result in Th2 inflammation-related asthma. However, in vitro studies have shown that the addition of PD-L2 to normal mice resulted in decreased T cell numbers and reduced production of Th2 cytokines. This suggests that the six reported cases of asthma following ICI administration might have had a pre-existing predisposition to asthma. Overall, the impact of PD-L2 on Th2 cytokine production may differ between individuals with and without a background of asthma, as indicated by the studies [75]. However, research on this mechanism is lacking, and further accumulation of cases and studies is necessary to better understand this mechanism.

### 3.8. Eosinophilic Fasciitis

Eosinophilic fasciitis (EF) is an inflammatory disease characterized by symmetrical swelling and hardening of the distal limbs. Laboratory examinations often show high levels of peripheral eosinophils, hypergammaglobulinemia, and elevated aldolase and erythrocyte sedimentation rates [81]. Additionally, skin biopsies extending to the fascia reveal thickening of the fascia with lymphoplasmacytic infiltration, followed by fibrosis of the interlobular septa [82]. The pathogenesis of EF suggests an increase in IL-2, IL-5, IL-10, and interferon-γ, which leads to eosinophilia and overexpression of immunoglobulins. The increase in eosinophils and IL-5, a type of Th2 cytokine, is associated with the pathogenesis of EF. Therefore, EF is also considered a Th2-type inflammatory disease [83]. 

To date, 14 cases of eosinophilic fasciitis have been reported following ICI administration [84,85,86]. In all 14 cases, oral corticosteroid therapy was required, and in more than half of the cases, oral methotrexate was also needed. However, there are no detailed reports on the mechanisms by which PD-1 inhibitors induce EF. Nevertheless, considering that EF is a Th2-type inflammatory disease closely related to IL-5 [82,83], it is possible that PD-1 inhibitor administration might enhance Th2-type inflammation due to the activation of PD-L2 functions, leading to EF development. Ultimately, further research and an accumulation of cases are needed to clarify the underlying mechanism of EF occurrence following ICI treatment.

### 3.9. Th2 Inflammatory Skin Diseases (PN, AD, BP, and EF)

Various Th2-type inflammatory skin diseases, including BP, AD, PN, and EF, have been reported following ICI administration. 

#### 3.9.1. Bullous Pemphigoid

Bullous pemphigoid (BP) is the most frequently occurring Th2-induced inflammatory disease, and a systematic review of BP induced by ICI therapy has already been published [87]. BP often develops during chemotherapy with anti-PD-1, anti-PD-L1, and anti-cytotoxic T lymphocyte-associated antigen 4 therapy. Approximately half of the cases of ICI-induced BP experience prodromal symptoms such as itching or nonspecific skin rashes. In most cases, ICI therapy was discontinued after BP onset, and oral corticosteroid therapy was administered [87]. Notably, Th2-related cytokines are implicated in the pathogenesis of BP, an autoimmune skin disease. Increased expression of Th2 cytokines such as IL-4, IL-5, IL-6, IL-10, and IL-13 has been observed in serum, skin biopsies, and blister fluid, indicating a close association with BP development [88]. Moreover, dupilumab, an IL-4 and IL-13 inhibitor, has shown promising results in preliminary clinical trials for BP treatment, further suggesting the involvement of Th2-type inflammation in BP [89].

#### 3.9.2. Eosinophilic Folliculitis

Rossi et al. reported a case of EF following PD-1 inhibitor administration [90]. This is the only reported case of scalp folliculitis following PD-1 inhibitor therapy. Similar to other eosinophilic diseases, it is possible that PD-1 inhibitor administration enhances Th2-associated inflammation through PD-L2 activation, contributing to the development of EF [36].

#### 3.9.3. Atopic Dermatitis

Tolino et al. reported a case of AD developing 11 months after administration of pembrolizumab in a patient with metastatic melanoma [91]. The patient’s skin symptoms improved with dupilumab, an IL-4/IL-13 inhibitor. Although reports of AD following PD-1 inhibitor administration are rare, the incidence of chronic eczema following ICI therapy is higher, suggesting that AD cases may be under-reported [29,30]. 

The pathophysiology of AD is complicated and diverse, involving barrier dysfunction, altered cell-mediated immune responses, IgE-mediated hypersensitivity, and environmental factors [92]. AD is a chronic pruritic immune-mediated inflammatory skin disease characterized by a Th2 immune response phenotype [93]. Th2 cytokines such as IL-4 and IL-13 are involved in its pathogenesis, and dupilumab, which inhibits these cytokines, has been effectively used in treating AD [93]. Similar to other diseases, PD-1 inhibitor administration may enhance Th2 inflammation through PD-L2 activation, leading to the production of IL-4 and IL-13 and the subsequent development of AD [36,93].

#### 3.9.4. Prurigo Nodularis

To date, two cases of PN following ICI administration have been reported, including our case. Fattore et al. reported a case of PN developing three months after PD-1 inhibitor administration in a 60-year-old woman [94]. In our case, a 51-year-old woman developed PN six months after starting nivolumab for lung adenocarcinoma. 

PN is a long-lasting inflammatory cutaneous disease characterized by intensely pruritic papules, pustules, and nodules being primarily distributed on the trunk and extremities [95]. Repetitive scratching exacerbates the condition, leading to the development of chronic lesions and worsening disease severity [95]. The lesional skin in PN patients exhibits infiltration by various immune cells, including eosinophils, neutrophils, macrophages, mast cells, and T cells, with a predominant Th2-type inflammatory pathway driven by CD4+ Th2 cells [40,96]. Additionally, increased levels of Th2 cytokines such as IL-4, IL-5, IL-10, IL-13, and IL-31 are also observed in the dermis of PN patients [40]. These cytokines bind to receptors on peripheral nerves, activating specific subsets of transient receptor potential cation channels, leading to the perception of itch and perpetuating the itch–scratch cycle [97]. Specifically, the pathogenesis of PN involves increased production of Th2-type cytokines, including IL-4, IL-13, and IL-31, which also drive the itch–scratch cycle [98]. While the mechanism by which ICI therapy induces PN has not been thoroughly discussed, it is hypothesized that PD-1 inhibitor administration may shift the immune response towards Th2-type inflammation through PD-L2 activation, contributing to the development of PN [36].

## 4. Discussion

Diseases induced by ICIs that are related to Th2 cytokines are mostly systemic eosinophilic diseases, such as EP, eosinophilic fasciitis, eosinophilic enteritis, eosinophilic granulomatosis with polyangiitis, eosinophilic sinusitis, and eosinophilic asthma, as mentioned above [40,41,42,43,44,45,46,47,48,49,50,51,52,53,54,55,56,57,58,59,60,61,62,63,64,65,66,67,68,69,70,71,72,73,74,75,76,77,78,79,80,81,82,83,84,85,86,87,88,89,90,91,92,93,94,95,96,97,98]. Skin diseases induced by ICIs that are related to Th2 cytokines include EF, PN, AD, and bullous pemphigoid. However, there have only been limited cases of EF, PN, and AD following ICI administration [90,91,94]. It is possible that various cases of Th2-type inflammatory diseases of the skin following ICI treatments may have been overlooked due to the lack of recognition of Th2 cytokines’ effects in this context. 

Theoretically, Th2 immunity is regulated by the PD-1–PD-L2 axis. PD-1 inhibitors may activate Th2 immunity by disrupting this axis, affecting not only internal organs but also the skin [36]. Previously, studies have shown that cutaneous dendritic cells possess PD-1 receptors and regulate Th2 cytokines. This suggests that PD-1 inhibitors could increase Th2 cytokine production by inhibiting dendritic cells [99,100]. However, most attention has been focused on the activation of Th1 and Th17 immunity through the PD-1–PD-L1 axis with PD-1 inhibitors, and the PD-1–PD-L2 axis related to Th2 immunity has not been widely discussed [36]. Notably, PD-L2 has a higher relative affinity for PD-1 compared to PD-L1, which may explain why PD-L1-related Th1 and Th17 cytokines are more readily activated when PD-1 inhibitors are administered [35]. However, there are no reports or research on why diseases involving the PD-1–PD-L2 axis are less common, necessitating further discussion.

The expression of PD-L2 is suggested to be driven by IL-4 [36]. In our case, the patient had no history of skin disease as an adult but had a history of AD in childhood. This may have resulted in a higher baseline production of IL-4. Therefore, it is possible that this was a factor leading to an increased expression of PD-L2, resulting in Th2-type inflammation and the development of PN. Research reports examining cytokines in the skin of patients with ICI-induced cutaneous symptoms have found increased production of Th2 cytokines such as IL-4, IL-5, and IL-13 in cases of urticaria, pruritus, and eczema [101]. Furthermore, the administration of dupilumab, which inhibits Th2 cytokines such as IL-4 and IL-13, has been effective in over 85% of patients with ICI-induced skin symptoms. This supports the hypothesis that ICIs can induce Th2 cytokine-related dermatitis [102]. Therefore, it is possible that many cases of eczema induced by ICI administration may actually be Th2-type inflammatory skin lesions, akin to AD, even though such inflammatory types are not observed in most cases. 

Gambichler et al. reported a case involving a 53-year-old male patient with disseminated Kaposi sarcoma and long-standing AD. The AD improved following treatment with the initiation of a PD-1 inhibitor [103]. On the other hand, Donato et al. reported a case involving a 63-year-old male with lung squamous cell carcinoma, asthma, and mild eosinophilia who developed allergic bronchopulmonary aspergillosis and marked eosinophilia after receiving pembrolizumab treatment. In their case, the patient’s eosinophil levels, which were initially mildly elevated, significantly increased, suggesting that the pre-existing Th2-type inflammation may have been exacerbated by the administration of the PD-1 inhibitor [63]. As demonstrated in these cases, the use of PD-1 inhibitors can yield different outcomes in patients with Th2 inflammatory diseases. Some cases show improvement in the original Th2-type inflammatory disease, while others exhibit exacerbation [63,103]. This discrepancy may depend on whether the PD-1–PD-L2 axis or the PD-1–PD-L1 axis is activated. In other words, stronger activation of the PD-1–PD-L1 axis may shift immunity from Th2 to Th1-type inflammation, leading to an improvement in pre-existing Th2-type inflammation. On the other hand, stronger activation of the PD-1–PD-L2 axis may exacerbate pre-existing Th2-type inflammation [36,63,103]. Nevertheless, these observations are currently limited to case reports, emphasizing the need for further research.

ICIs are also known to induce tissue-resident memory T cells [37]. In our case, the persistent itching after discontinuation of nivolumab could be due to the influence of these memory T cells induced by the ICI administration in addition to continuous scratching associated with itching. Presently, the itching in this patient is being managed with oral antihistamines and topical steroids, and this treatment will continue. However, if symptoms worsen or if the patient requests it, we plan to administer dupilumab based on previous reports [101,102]. 

Reports on ICI-related Th2 inflammatory diseases, specifically those involving IL-4 and IL-13 cytokines, are very limited, and there is a lack of literature discussing Th2 inflammation induced by ICIs. Overall, the relationships among ICIs, PD-1, PD-L2, dendritic cells, and Th2 inflammation discussed in this report remain hypothetical. Moreover, the exact mechanisms by which ICIs contribute to the development of Th2-mediated diseases are still unclear, and further accumulation of cases and research is needed to understand these mechanisms.

## Figures and Tables

**Figure 1 biomedicines-12-01886-f001:**
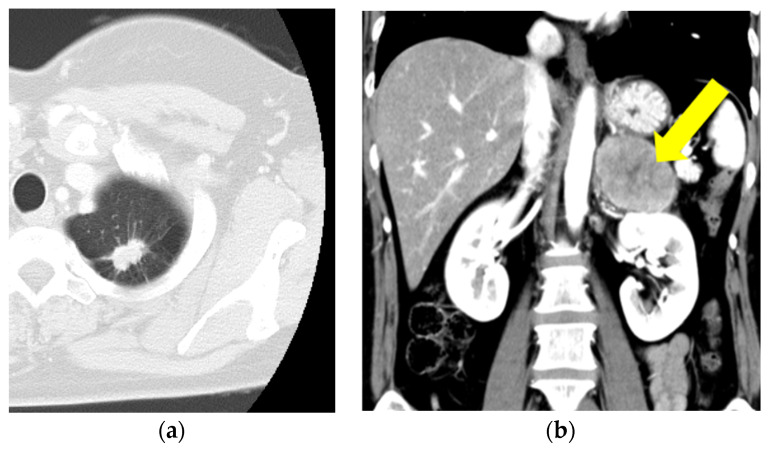
(**a**,**b**) Computed tomography reveals lung adenocarcinoma (**a**) and adrenal metastasis (**b**) (yellow arrow).

**Figure 2 biomedicines-12-01886-f002:**
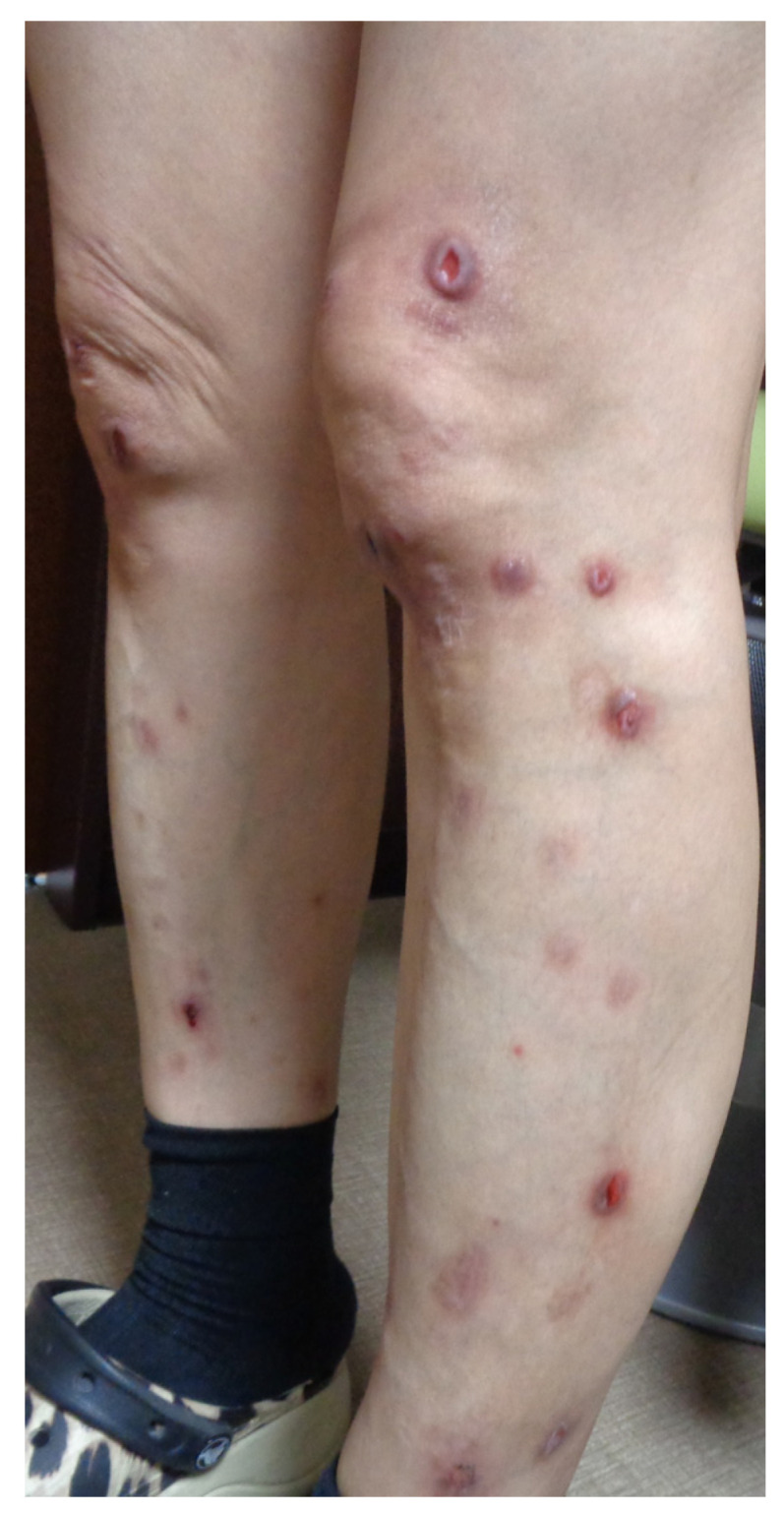
Physical examination reveals multiple prurigo on extremities, some with ero sions and ulcers in the center and some with hyperpigmentation even after nivolumab was discontinued.

**Figure 3 biomedicines-12-01886-f003:**
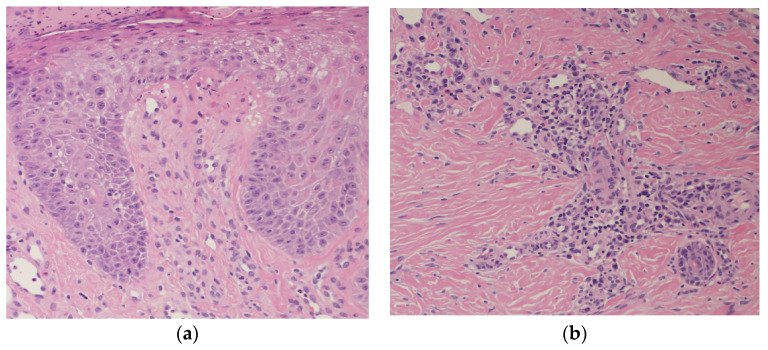
(**a**,**b**) Histopathological examination from nodular prurigo reveals epidermal thickening, liquefaction degeneration, spongiotic dermatitis ((**a**) hematoxylin and eosin stain (HE) ×100), and inflammatory cellular infiltration around vessels in the dermis ((**b**) HE ×100).

## Data Availability

The original contributions presented in the study are included in the article, so further inquiries can be directed to the corresponding author.

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
