# Peer review of "Review of T Helper 2-Type Inflammatory Diseases Following Immune Checkpoint Inhibitor Treatment"

_biomedicines, 2024, doi:10.3390/biomedicines12081886_

Round 1

Reviewer 1 Report

Comments and Suggestions for Authors

This is excellent review on Th2 mediated inflammatory diseases induced by immune checkpoints inhibitors (ICIs). My only suggestion is to edit the first statement of Introduction that implies that tumor cells do something deliberately “to avoid attacks from the immune system”. Although this is a common theme in tumor immunology that works well for the lay audience, in reality, tumor cells make no deliberate decisions of any kind but simply react to the local and systemic cues based on their genetically embedded programs.  One could expand on explaining why tumor microenvironment is immunosuppressive and what kind of preceding circuits between the tumor mass and immune regulators in bone marrow and spleen are involved in establishment of this pattern. This would help to better understand the interplay of Th1/Th2 regulation and the impact of ICIs described in the main body of the review.

Author Response

Comments:

"This is excellent review on Th2 mediated inflammatory diseases induced by immune checkpoints inhibitors (ICIs). My only suggestion is to edit the first statement of Introduction that implies that tumor cells do something deliberately “to avoid attacks from the immune system”. Although this is a common theme in tumor immunology that works well for the lay audience, in reality, tumor cells make no deliberate decisions of any kind but simply react to the local and systemic cues based on their genetically embedded programs.  One could expand on explaining why tumor microenvironment is immunosuppressive and what kind of preceding circuits between the tumor mass and immune regulators in bone marrow and spleen are involved in establishment of this pattern. This would help to better understand the interplay of Th1/Th2 regulation and the impact of ICIs described in the main body of the review."

Reponse:

Thank you very much for your kind and thorough review.

As you pointed out, the expression suggesting that tumor cells are deliberately evading immune surveillance has been revised to reflect that tumor cells are programmed to escape immune responses. These points have been added to the Introduction section, lines 39-61 as below.

"The tumor microenvironment (TME) is inherently immunosuppressive due to the coordinated actions of tumor cells and their surrounding stroma, which collectively create conditions favorable for tumor progression and immune evasion [1-5]. Tumor cells frequently overexpress immune checkpoint molecules, leading to the inhibition of T cell activation and a subsequent reduction in immune responses [1]. Additionally, the accumulation of immunosuppressive cells such as regulatory T cells (Tregs) and myeloid-derived suppressor cells (MDSCs) within the TME further impedes effective immune responses against tumor cells [2]. Tumor cells also secrete immunosuppressive cytokines, including IL (IL)-10 and transforming growth factor-beta (TGF-β), which further inhibit T cell function and weaken anti-tumor immunity [3]. The TME is often marked by hypoxia and acidic conditions, which additionally compromise immune cell function [4]. Furthermore, tumor cells can downregulate the expression of major histocompatibility complex (MHC) class I molecules, making it more challenging for T cells to recognize and attack tumor antigens [5]. These factors collectively establish an immunosuppressive milieu that allows tumors to evade immune surveillance [1-5]. Additionally, tumor cells can send immunosuppressive signals to the bone marrow and spleen, which are crucial for the generation and maintenance of Tregs and the production of cytokines and chemokines, thereby perpetuating an immunosuppressive state [2,6-7]. The intricate interplay between tumor cells, the TME, and these hematopoietic organs ensures the sustenance of an immunosuppressive environment that supports tumor growth [2,6-7]. Consequently, immune checkpoints—central to maintaining this immunosuppressive balance by inhibiting T cell activation—are a primary focus of therapeutic strategies [1]."

Tumor cells highly express immune checkpoints molecules, increasingly produce immunosuppressive cytokines, and downregulate MHC class I molecules. Tumor microenvironment produces immunosuppressive cells and creates specific conditions such as hypoxia and acidity. Consequently, tumor cells and tumor microenvironment can maintain immunosuppressive condition, leading to evasion from immune surveillance and tumor growth.

Additionally, tumor cells send immunosuppressive signals to the bone marrow and spleen, which are both involved in producing immunosuppressive cells and cytokines, thereby playing a crucial role in maintaining the immunosuppressive state of the tumor and its microenvironment.

These points have been added to the Introduction section, lines 39-61.

Moreover, the portions of the text that overlap with past literature have been revised for expression only, while the content remains the same. These sections are clearly indicated with underlined black text.

Thank you very much.

Reviewer 2 Report

Comments and Suggestions for Authors

I have carefully and with great interest read the manuscript of Yoshihito Mima et al. entitled "Review of T helper 2-type inflammatory diseases following immune checkpoint inhibitor treatment".

The study raises the issue of exposure to immune checkpoints and describes the signaling mechanisms that occur during this exposure.

The topic of the paper is highly relevant, as understanding the mechanisms of the body's response when immunosuppressive drugs are used is very important and will provide a deeper understanding of the nature of cancer processes, ways to influence them and methods of avoiding negative immune effects.

The specific case of a rare immune response of a patient on immunosuppressive treatment is reviewed, and the possible morbidities of this treatment are summarized and systematized.

In my opinion, the work is of a high scientific standard, written in an accessible and understandable language and does not require improvements in the description of the methodology.

The conclusions of the paper fully reflect the results described and summarize the research objective.

The authors have done a lot of work to generalize previously published materials on the research topic.

1) The terms in vivo, in vitro, et al. and the like should be italicized

2) It would be good to introduce a list of abbreviations, as there are quite a lot of them and it is difficult to search for them in the long text.

I believe that after eliminating the shortcomings, the work can be published in the journal Biomedicines.

Author Response

Comments:

I have carefully and with great interest read the manuscript of Yoshihito Mima et al. entitled "Review of T helper 2-type inflammatory diseases following immune checkpoint inhibitor treatment".

The study raises the issue of exposure to immune checkpoints and describes the signaling mechanisms that occur during this exposure.

The topic of the paper is highly relevant, as understanding the mechanisms of the body's response when immunosuppressive drugs are used is very important and will provide a deeper understanding of the nature of cancer processes, ways to influence them and methods of avoiding negative immune effects.

The specific case of a rare immune response of a patient on immunosuppressive treatment is reviewed, and the possible morbidities of this treatment are summarized and systematized.

In my opinion, the work is of a high scientific standard, written in an accessible and understandable language and does not require improvements in the description of the methodology.

The conclusions of the paper fully reflect the results described and summarize the research objective.

The authors have done a lot of work to generalize previously published materials on the research topic.

1) The terms in vivo, in vitro, et al. and the like should be italicized

2) It would be good to introduce a list of abbreviations, as there are quite a lot of them and it is difficult to search for them in the long text.

I believe that after eliminating the shortcomings, the work can be published in the journal Biomedicines.

Response:

Thank you very much for your kind and thorough review.

 We have made the following revisions based on your suggestions:

1) All terms such as "in vivo," "in vitro," and "et al." have been changed to italicized text.

2) A list of abbreviations has been added immediately after the Discussion section in the main text, line 573-584.

"List of abbreviations:

TME: tumor microenvironment; Tregs: regulatory T-cell; MDSCs: myeloid-derived suppressor cells; IL: interleukin; TGF-β: transforming growth factor-beta; MHC: major histocompatibility complex PD-1: programmed death 1; PD-L1: programmed death ligand 1; CTLA-4: cytotoxic T-lymphocyte associated antigen 4; LAG-3: lymphocyte activation gene3 ; TCR: T cell receptor; IFN-γ: interferon-gamma; APC: antigen-presenting cells; TIM3: T cell immunoglobulin and mucin-domain containing 3; TIGIT: T cell immunoreceptor with Ig and ITIM domains; VISTA: V-domain Ig suppressor of T-cell activation; ICIs: immune checkpoint inhibitors; irAEs: immune-related adverse events; Th: T helper; AD: atopic dermatitis; PN: prurigo nodularis; HE: Hematoxylin and eosin stain; Ig: immunoglobulin; EP: eosinophilic pneumonia; EF: eosinophilic folliculitis; ABPA: allergic bronchopulmonary aspergillosis; EGPA: eosinophilic granulomatosis with polyangiitis; BP: bullous pemphigoid"

Moreover, the portions of the text that overlap with past literature have been revised for expression only, while the content remains the same. These sections are clearly indicated with underlined black text.

Thank you very much.